# Feasibility of a Virtual-Reality-Enabled At-Home Telerehabilitation Program for Stroke Survivors: A Case Study

**DOI:** 10.3390/jpm13081230

**Published:** 2023-08-03

**Authors:** Mohamed-Amine Choukou, Elizabeth He, Kelly Moslenko

**Affiliations:** 1Department of Occupational Therapy, Rady Faculty of Health Sciences, University of Manitoba, Winnipeg, MB R3E 0T6, Canada; hee@myumanitoba.ca (E.H.); kellymoslenko@gmail.com (K.M.); 2Centre on Aging, University of Manitoba, Winnipeg, MB R3T 2N2, Canada

**Keywords:** remote care, stroke, technology, rural

## Abstract

Stroke rehabilitation is a lengthy procedure that is necessary for stroke recovery. However, stroke rehabilitation may not be readily available for patients who live rurally due to barriers such as transportation and expenses. This shortage in wearable technology, in turn, causes health disparity among the rural population, which was exacerbated by the COVID-19 pandemic restrictions. Telerehabilitation (TR) is a potential solution for stroke rehabilitation in rural areas. This one-case study aimed to examine the feasibility and safety of a technology-enabled at-home TR program for stroke survivors living in a rural area in Canada. A VR setup was installed successfully in the home of our participant. A tablet was also supplied for the TR program. Each program consisted of 24 sessions to be completed over a 12-week period. Our participant was assessed on day one using the Fugl-Meyer assessment, the Modified Ashworth Scale, the 10 m walk test, and the Mini-Mental State Exam. Three questionnaires were also completed, including the Motor Activity Log (MAL), the Stroke Index Scale (SIS), and the Treatment Self-Regulation Questionnaire. These assessments were completed thrice, on day 1, at week 6, and at week 12. The participant found the tablet and its accompanying exercises easy to use, with a few limitations. The participant found the VR system more challenging to manage independently as a lack of comfortability, the visual contrast during the first trials, and certain technical aspects of the technology created several functional barriers. Although some limitations with the technology were noted, this case study indicates that telerehabilitation is feasible under certain circumstances when used in conjunction with traditional rehabilitation services.

## 1. Introduction

Stroke rehabilitation is a lengthy and intensive process that usually occurs over several months, with the first three to six months being a crucial period for recovery [1,2]. Repetitive and task-oriented training has shown to be a necessary component of stroke rehabilitation to produce the desired neuroplastic changes within the brain in order to improve motor deficits and promote functional recovery [3]. Traditional rehabilitation is tedious, costly, labor-intensive, and requires strict adherence, with most traditional stroke therapy sessions lacking the ability to provide the sufficient repetitions required for critical neuroplastic changes to be made [3]. Furthermore, logistical, financial, and geographical barriers to such specialized rehabilitation programs may prevent stroke survivors from accessing adequate and necessary care within the critical first three-to-six-month period [4]. Location is a significant barrier for those who live rurally. For example, many patients require assistance from others for transportation or must pay out of pocket for transportation services. Due to the distance and inconvenience, patients are deterred from attending their rehab appointments [5]. The quality of roads and harsh winter climates make it even more complicated to seek healthcare services, as most stroke clinics are located in urban areas [6]. These issues are also exacerbated for underserved populations (e.g., people living in remote areas or Indigenous populations) that already face many health inequities such as costs, distance, racism, substandard care, and discrimination [7], and one could expect this situation to worsen under the COVID-19 pandemic restrictions. Barriers also exist regarding the financial aspect of stroke therapy, both for the Canadian economy and for the individual patient. The cost of those living with strokes is CAD one billion annually [6]. Even with such a significant expenditure, the Canadian healthcare system still finds itself understaffed and limited in its resources, making it difficult for patients with stroke to find adequate care. Stroke survivors are therefore required to pay the expenses of travelling to rehabilitation centers while taking time off work and/or asking a caregiver to take the same time off to receive these services [5].

The intensity of rehabilitation and the many barriers individuals may face in accessing these programs have created a need for accessible outpatient rehabilitation services that can be delivered safely and effectively to ensure equitable recovery for all stroke survivors. Telerehabilitation (TR) is an alternative to conventional in-person rehabilitation that uses information and communication technologies to deliver rehabilitation services [8]. The expanding literature on TR for stroke has indicated its effectiveness for various recovery domains, including the motor, cognitive, and affective domains [4,8]. The technology varies across the literature, including telephones, smartphones, computers, video conferencing, and virtual reality (i.e., immersive, semi-immersive, and non-immersive modes) [9,10].

This project introduces the “Active at Home Post-Stroke Program” (or Home PSP) (Figure 1), a novel program developed by Dr. Amine Choukou that combines virtual-reality-enabled cognitive training with tablet-based physical activity. Although this TR program can be administered as a standalone program, it is proposed in this study as an adjunctive therapy to conventional rehabilitation services, with the primary goal of determining the program’s feasibility and safety. The study is designed with the physical and cognitive rehabilitation of stroke survivors as the primary aim of the rehabilitation program. The Home PSP program is delivered to participants in two ways. One is an interactive mobile app that was also developed by Dr. Amine Choukou and is installed on an Android tablet. The other delivery method uses virtual reality technology, with participants taking part in physical and cognitive training sessions alternately throughout the program. During the physical rehabilitation, a virtual coach delivers the program in a three-month progressive exercise program. This program includes resistance training, balance training, and stretching. The cognitive aspect of the training is a simulated VR environment, with activities for participants to complete. The activities resemble the scenarios we encounter during daily life inside the home. The expectations of Home PSP are that it will guide stroke survivors toward regaining increased levels of physical and cognitive abilities following a stroke. Depending on the individual, physical and cognitive performance are variably affected following a stroke. Home PSP is designed with this consideration as a top priority.

This study aims to determine the feasibility and safety of a technology-enabled at-home telerehabilitation program for stroke survivors living in rural areas. The objectives were to evaluate the feasibility and safety of introducing virtual rehabilitation into the home of a stroke survivor, evaluate a participant’s satisfaction with the technology and note any marked improvements in physical function, self-reported quality of life, and motivation, and determine potential barriers and limitations to using the technology to problem solve and correct for future use.

## 2. Materials and Methods

### 2.1. Intervention Protocol

This is a functional rehabilitation program based on immersive VR simulations of tasks related to activities of daily living and a tablet program that provides “follow along” videos of different exercises that target gross and fine motor skills and strength and range of motion. There were 48 sessions for the VR and tablet (24 sessions each) across a total of 12 weeks. Each week, the participant would complete two sessions for each technology, resulting in four sessions each week. A recent systematic review with a meta-analysis recommends that the physical training program should last at least six weeks, with a frequency of three sessions per week and a session duration of at least 30 min [9]. A randomized controlled trial by Bo et al. (2019) tested a combination of physical and cognitive training for 36 sessions, three days per week, for 12 weeks in a medical rehabilitation centre [10]. Our study was conducted remotely and was semi-supervised. In agreement with the literature and taking into account the exploratory nature of our program management, we suggest administering the program over 4 sessions a week, alternating 2 cognitive training sessions and 2 physical training sessions over 12 weeks. Additionally, to increase the intensity of the program over time while keeping safety as the foremost priority, the complexity of each session gradually increased, and the total number of tasks also increased.

#### 2.1.1. Physical Training Program (A Tablet App)

Each participant was given a tablet with the tablet app downloaded to it. Participants logged in and completed the session by completing each exercise. Each session consisted of exercises targeting the upper and lower body and focusing on gross and fine motor skills. The program involves a blend of resistance, balance, and stretching training. Users can complete activities in the order they prefer within two consecutive sessions for ease, comfort, and motivation. They cannot override the second session until the two sessions have been completed. Sessions increase in terms of the complexity and duration of the exercise as the program continues. Activities are to be performed in front of the tablet in a standing and/or seated position to allow the user to do what they can safely and comfortably every time they practice. Help from the family caregiver is not expected, as users can operate the tablet with an unimpaired hand. After each exercise, a questionnaire appears, asking for feedback about how the activity went. The information is then automatically emailed to the principal investigator (no action is required from the tablet user).

#### 2.1.2. Neurocognitive Training Program (A Virtual Reality Environment)

The HTC Vive Pro system was used to run the VR program. The setup consists of a headset, two hand-held controllers, and two base stations on a tripod. The VR system was set up in the participant’s home for seated and standing use. The project’s principal investigator designed the environment and the cognitive activities based on a virtual apartment that replicates a newly developed ambient assisted living facility [11]. The activities resemble scenarios we encounter daily inside the home. Figure 2 presents some examples of activities and interactions in the VR environment. There are five total training modules with a total of 24 sessions. Each session included several household tasks, requiring the participant to move around the virtual home using the left and right hand-held controllers to interact with the spaces and objects. As the sessions progressed, the tasks became more complex, necessitating more steps to complete them. The activities are to be performed in front of the computer in a seated position to allow the user to do what they can safely and comfortably. Help from the family caregiver is needed to setup the VR station (PC, headset, controllers) and is recommended during the practice to increase the participant’s sense of safety. The data log is saved and transmitted to a dedicated server owned and secured by the university immediately following each practice. The first four modules contain five sessions each and the last includes four. Each module has its own objectives:Module 1: Familiarity with the Environment and Technology

Module 1 aims to instruct the user how to manipulate the material and interact with the VR environment in an incremental way (teleportation from one point to another, exploration, zooming in and out, holding objects, clicking on buttons, etc.). This module instills the participant’s technical skills and confidence in using VR to perform basic tasks. The activities involve putting on the helmet, learning the basic commands, and getting comfortable moving with the technology.

Module 2: Task Management and Prioritization

The second module has the same structure as the first module. It focuses on task management, the way the user prioritizes actions based on common sense and preferences and the management of the interruptions of a task in progress. The activities are added progressively so that the client can complete an entire day by the module’s last session. The goal is to manage tasks, identify priorities, and sometimes even interrupt a task in progress to meet an immediate need, such as answering the phone.

Module 3: Financial Management

Module 3 aims to enhance budget awareness and money-management abilities. A total of CAD 100 is proposed for the five sessions of this module. Throughout each session, the user will be required to make financial judgements while staying under CAD 100. If they do, they must be able to explain their financial decisions in the questionnaire at the end of the module, because every financial decision must be considered as part of the whole module’s financial management plan. For example, if the user has no laundry detergent left and must pick between three varieties of detergent, they must remember the cost and deduct it from the overall budget for this module (CAD 100). This module asks the customer to recall the expenses incurred during the five sessions. Clients might create their own memorization approach by writing down the costs incurred shortly before the session ends. Clients are free to use whatever method they feel most at ease with.

Module 4: Tasks and Financial Management

The fourth module continues the format of the previous modules but attempts to provide the client more autonomy to carry out the tasks due to the practice they have completed. This module prepares the client for the final module by improving the abilities needed to understand and manage contingencies and adapt to varied scenarios. In the fourth module, activities are gradually introduced so that, by the end of the module, the user can complete an entire day. The sessions are more sophisticated, combining interpersonal and financial issues. Furthermore, the scripts are done away with to increase task learning and retention because the actions have already been covered in earlier sections.

Module 5: Full Day

The fifth module aims to move closer to the reality of daily activities. It consolidates the learning gained in earlier modules, allowing the evaluator to determine whether the client can respond appropriately to complex sessions. The user encounters complex scenarios in this module, as each of the four sessions entails dealing with unexpected everyday life situations and financial situations for an entire day. Therefore, the evaluator will understand the acquired skills and the underpinning cognitive abilities.

**Figure 2 jpm-13-01230-f002:**
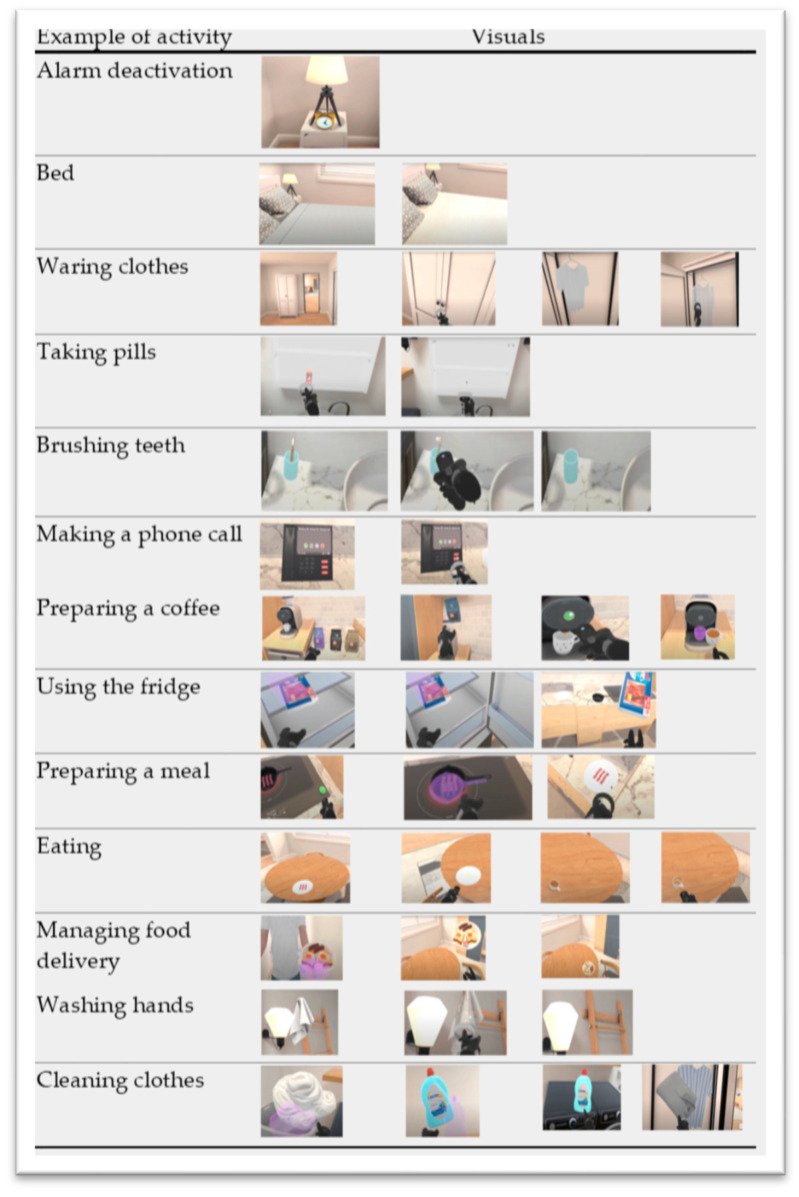
Examples of activities and interactions in the VR environment.

### 2.2. Study Variables

The protocol was approved by the University of Manitoba Research Ethics Board (REB Registry Number: H2020:069 (HS23654)). The participants signed a consent form electronically before the start of the study. Following participant enrollment, a battery of assessments was performed by a physiotherapist on day 1 (in our laboratory), followed by re-assessments at home during weeks 6 and 12. These assessments included the Fugl-Meyer assessment (FMA) [12], the Modified Ashworth Scale (MAS), the 10 Meter Walk Test (10MWT) [13], the Mini-Mental Status Exam (MMSE) [14], and the Borg rating of perceived exertion (BORG) [15]. The purpose of these assessments was to evaluate the participant’s physical and cognitive function and their perceived exertion during rehabilitation exercises.

In addition to these assessments, the Motor Activity Log (MAL) [16], Stroke Impact Scale (SIS) [17], and Treatment Self-Regulation Questionnaire (TSRQ) [18] were provided to each participant in week 1, followed by a re-assessment in week 7 and week 12. These instruments were chosen to assess the participant’s functional ability, quality of life, and self-regulation in their rehabilitation process. The participant had weekly informal check-ins to discuss their progress over the past week.

To gain insight into the family caregiver’s perspective, an email was sent to them after the study completion to enquire about their “attitudes and perceptions towards the home-based rehabilitation program and their experience with the research project” (the main question of the email). The responses received were exhaustive and highly organized. We analyzed the caregiver opinions verbatim, using an inductive creation-of-themes method to classify and understand the content shared.

### 2.3. Participant Inclusion and Experimental Setup

A participant contacted the principal investigator by email after attending a public presentation. The participant was screened to determine their eligibility for the program according to the following inclusion criteria:Mild stroke, onset over six months before the initiation of the program;Absence of any additional neurological or psychiatric disorders or orthostatic hypotension;Able to walk and move the impaired upper limb;Willing to participate;Living outside the city with access to the internet at home;Living with someone who can help in the operation of a VR system.

Once eligibility to participate in our study was confirmed, the participant was invited to an orientation session, which included an introduction to the VR and tablet program. The tablet running the physical training program was handed out to the participant on the same day and instruction on how to use it was given. The VR platform, comprising a gaming computer and a VR system, was installed in each participant’s home in the following week. The participant and their caregiver received a training session at home on how to use the VR.

### 2.4. Participant

Our participant is a 55-year-old woman who experienced a stroke in March of 2021. The participant lives in a house a 1.5–2-h drive from the main city in a community of fewer than 15,000 inhabitants. Their home is equipped with reliable internet and has a large and safe space, sufficient to engage in our program. The stroke affected her right upper and lower extremities, increasing the tone mainly in her arm and hand. The stroke also affected her speech, which resulted in aphasia. After the stroke, the participant attended two months of stroke rehabilitation, where an interdisciplinary team worked with her. This team consisted of occupational therapists, physiotherapists, and speech-language pathologists. After discharge, she continued seeking out therapy regularly for her physical functional needs and speech training. At the time of the study, the participant was regularly seeing a private practice physiotherapist, an occupational therapist, a speech and language pathologist, and a personal trainer.

In the day 1 assessments, our participant received a score of 30/30 on the MMSE, which is within the normal range [14]. On the FMA, she scored 38/66, which indicates “limited capacity” [19]. On the MAS, the participant scored a “2”, which indicates a “more marked increase in muscle tone through most of the range of motion, but affected parts easily moved” on each area of upper extremity flexion, extension, rotation, and supination. For the 10MWT, the participant’s speed for the self-determined velocity was 7.68 s. For the BORG, she received an overall rating of “2”, which indicates “Light activity: Feels like you can maintain for hours, easy to breathe and easy to maintain a conversation” for the 10MWT. The participant was observed to ambulate with an unbalanced gait due to her right side being affected, and she was able to raise her arm toward objects without being able to catch or manipulate the objects.

## 3. Results

Weekly check-ins were performed through informal phone interviews. The purpose of the interviews was to discuss generally what went well and what did not. Additionally, observations and data from the actual performance were automatically uploaded to the principal investigator’s server via the programs.

During the first check-in, the participant noted some general sickness that prevented them from being able to continue the first few sessions of the program. The MAL, SIS, and TSRQ were completed. In the TSRQ, the participant reported that it was important for them to exercise and get well soon. The participant also indicated strongly that they would feel guilty if they did not follow through with the program. On the SIS, the participant self-scored an overall value of 60/100 regarding their perceived level of recovery.

The participant reported that they were able to complete the first VR session, but they needed the assistance of their caregiver to press the action button located on the top of the controller. Because of the high tone in the right hand, the participant was not able to press the action button independently. The ergonomics of the controller were not well suited to this particular situation.

In the third week, the participant did not complete any sessions due to their unavailability during this time for personal reasons.

In the fourth week, no further VR sessions were completed as the caregiver was not available to assist with the right-hand controller. One additional tablet session was completed. The participant noted that it was significantly more difficult due to the inclusion of exercises that required them to move their arm above the shoulder. There was no noted pain during this movement. The participant was also interested in modifying the controller to meet the practice needs. After discussions with the research team, the family caregiver (partner) attached a brace with adhesive Velcro to attempt to make it more functional and a plate to neutralize undesired thumb movement (Figure 3). The participant also preferred to increase the number of tablet sessions per week, which was approved by the research team.

In week five, the participant experienced difficulty with fine motor tasks as the sessions continued. The participant did not experience difficulty with tasks involving gross motor skills. The participant did not continue with any VR sessions due to issues with the controller. The participant also noted that they experienced difficulty initiating the sessions due to motivation; however, they did not experience any difficulty completing the session. The participant suggested that a research team member send motivational emails daily, and the research team agreed.

In week six, the participant noted that their caregiver was able to modify the controller to make it more comfortable to wear and easier to maneuver. However, clicking the button for actions still required help from the caregiver. The participant also completed one additional tablet session. They noted that it was easier as there were no exercises involving fine motor tasks. The motivational and prompting emails were helpful but the participant was not able to continue with many tablet sessions due to their own schedule becoming busier, causing fatigue.

In week seven, the three questionnaires were re-administered. For the TSRQ, most of the scoring was the same, except for a few questions that asked whether participant would feel bad if they did not complete the program. This shifted to a “2”, which indicates “not really”. This is different to week 1, where they had answered “4”, which indicates strongly that they would feel guilty. Regarding the SIS, the participant self-scored higher on the overall recovery level with a score of 70/100. Tasks that were originally scored as “very difficult” to perform were now scored as “could not do at all”. This is consistent with certain MAL tasks compared to the first time. There was an increase in the number of tasks that scored a “0” on the amount scale, even though they were scored higher on day 1. Many of the tasks that shifted to a “0” were tasks that were focused on fine motor skills, such as using a doorknob or turning on the faucet. Instead, the participant used their unaffected arm and hand for compensation.

### 3.1. Week Six Assessment

The following results were measured in week six.

On the 10MWT, the average self-selected velocity was 6.15 s, whereas the fast velocity was 5.9 s, indicating an increase in the participant’s walking speed since day 1. Her score on the Borg scale of perceived exertion was “4”, which indicates “moderate activity: feels like you can exercise for hours, breathing heavily but can hold a short conversation” for the 10MWT. On the MAS, the elbow flexors/extensors, shoulder flexors, wrist, and finger extensors and flexors were reassessed and the participant’s score for each subset was “1”, indicating a reduction in their tone. A score of “1” indicates a “slight increase in muscle tone, manifested by a catch and release or by minimal resistance at the end of the range of motion when the affected parts are moved in flexion or extension”. For the MMSE, the participant’s score was 24/30, which is a significant reduction from day 1. This score suggests mild dementia. On the FMA, the score increased to 40/66 from day 1.

Our participant completed eight home training sessions for the VR within six weeks and five tablet training sessions within six weeks. The participant was not able to complete the recommended two sessions a week for each program due to illness. During the first six weeks, the participant could not work on sessions for two weeks. Through informal interviews, other circumstances were brought up, such as the need for a caregiver to complete the VR with the participant, which meant that they both had to be available and well. Motivation also appeared to have an influence on the number of tasks completed per week. The participant reported that being self-directed in therapy posed a challenge because there was no one physically there to keep them accountable.

### 3.2. Week 12 Assessment

The following results were measured after the last session of week 12.

On the 10MWT, the average self-selected velocity was 6.2 s, whereas the fast velocity was 4.1 s, showing an increase in the participant’s walking speed since week six. Her score on the Borg scale of perceived exertion was “4”, which indicates “moderate activity: feels like you can exercise for hours, breathing heavily but can hold a short conversation” for the 10MWT. On the MAS, the elbow flexors/extensors, shoulder flexors, and wrist flexors/extensors were assessed and given a score of 1, indicating a reduction in their tone from day 1. A “1” indicates a “slight increase in muscle tone, manifested by a catch and release or by minimal resistance at the end of the range of motion when the affected parts are moved in flexion or extension”. However, the finger extensors/flexors and, wrist extensors, and thumb abductors were reassessed, and the participant’s score for each subset was “0”, indicating “No increase in muscle tone”. For the MMSE, the participant’s score was 24/30, which is the same as that recorded in week six but still a significant reduction from day one. This score suggests mild dementia. The FMA score had increased creased from 40/66 to 51/66 since Week 6.

Our participant completed 16 home training sessions for the VR within 6 weeks and 10 tablet training sessions within 12 weeks. The participant was not able to complete the recommended two sessions a week for each program due to illness and managed as best as they could. Motivation also appeared to have an influence on the number of tasks completed per week. The participant reported that being self-directed in therapy posed a challenge because there was no one physically there to keep them accountable. The participant did not fall during any of the practice sessions.

### 3.3. Family Caregiver’s Perceptions of Home PSP

We asked the family caregiver to share their opinions about their situation, their attitudes and perceptions toward the home-based rehabilitation program, and their experience with the current research project. The following themes emerged from the extensive essay they (the husband) shared by email. The caregiver felt devastated by the new situation, mentioning that “*one of the biggest challenges is dealing with the swiftness that this change has on [their] life. In [their] case, [his] wife, best friend, and confidant of 35 years was now lying in the hospital paralyzed and unable to speak*”*…* and crying “*…all the time and seemingly for no reason at times*”*…* while he is “*…running around making plans and decisions on [his] own. Something that [the couple] always did together*”. He also mentioned feeling alone in this new struggle without any external support, since the stroke “*kicked off in the blink of an eye and totally without warning*”, but he emphasized they are fortunate to have a comfortable financial situation: “*I am fortunate that we have good insurance coverage, and I was able to close my business without worrying about finances or outside commitments. I can’t imagine what this would be like while still trying to hold down a job and put bread on the table*”.

We also asked questions regarding access to care and systemic barriers. The family caregiver noted difficulties in accessing resources and care in a healthcare system that “*needs to catch up*”, according to the caregiver. “*Looking at this from the outside, it has been a challenging road that we have travelled in that the local hospital lacks the resources to give any meaningful aftercare or rehab to Stroke survivors*”. He emphasized that the situation is worse for someone living in a rural area where hospitals “*need to be updated to enable constructive and easily accessible treatment and rehabilitation*” and lack quality programs, according to the caregiver: “*A visit to the Occupational Therapy department of your local hospital once every two months is not good enough and the exercises prescribed are not always appropriate*”. The family caregiver also shared that he felt left behind by the entire health system, which is not prepared to care for patients in the city, almost absent in rural settings, heavily bureaucratic, and more or less unprepared to support the family caregiver. This creates, according to the caregiver, “*a myriad of issues for the spouse or family member who is thrust into the role of caregiver*” without any support or accompaniment:
*“You are being asked to do things you have never prepared for. Both in the physical aspect of caring for someone with disabilities and now non-functioning body parts and systems and the emotional side of mourning the loss of your spouse as you know them and your life as you know it.” … “The list of issues begins with having to deal with and navigate your way around hospitals, surgeons, physicians, nurses, travel, and accommodation. These are just a few things that the home caregiver requires within the first few weeks of their partner having a stroke.”*

The unpreparedness of healthcare professionals also emerged as a central theme. According to the family caregiver, who is not a healthcare professional, “*Training of Physiotherapists, Occupational Therapists, Speech Therapists, Primary Care Physicians and general staff is sorely lacking. Physiotherapists are trained to treat muscles, tendons and joints. Occupational Therapists are there to get patients back to work and make adaptations to allow for normal day-to-day activities, while Primary Care Physicians are not trained in the care of post stroke patients.”* … *“Family Physicians and health care workers, in general, need to be trained and informed of procedures and resources available to stroke survivors and their families*”.

## 4. Discussion

This study aimed to determine the feasibility and safety of a technology-enabled at-home telerehabilitation program administered remotely as an adjunctive therapy to conventional rehabilitation services for a stroke survivor living rurally in Manitoba (Canada). The program was tested with one participant with a mild stroke who was already able to walk and move the impaired upper limb and does not have any additional neurological or psychiatric disorders or orthostatic hypotension. The participant readily accepted the technology at home and found the tablet exercises enjoyable and easy to follow. The first hurdles in implementing our program in a rural setting were distance and access to the internet, required for the technology to transfer data to the researchers and for the family to interact with the researchers. The researchers were able to travel back and forth in a single day to install the device or conduct an evaluation session; thus, the distance was not an issue. There was no problem at all in communication because the household had internet access. However, this would have been an issue in other contexts where the internet is unavailable, in which case the data recorded by the two devices would not be shared, and the communication between the family and researchers would be more problematic or elementary, without any file-sharing or videoconferencing.

Overall, our preliminary results show that the VR environment effectively engages patients in telerehabilitation and is a safe virtual space for stroke survivors to regain neurocognitive skills. Our results are in agreement with the literature on less advanced VR systems offering gamified upper-limb training sessions [20,21,22,23]. However, our study is the first to introduce VR to practice daily living activities in an environment replicating a real-life smart apartment. Moreover, evidence has suggested the potential for motion sickness or cybersickness to function as a prominent barrier when utilizing head-mounted VR devices [24], the side effects of which can include dizziness, headaches, disorientation, and fatigue, which may limit the usability of VR; these were not side effects seen in this study. In our case, a lack of functional ability in the affected extremities created the most significant barrier for our participant to independently engage with the VR technology, which led to an unanticipated dependence on caregiver support. Even with the added assistance from the caregiver, the participant still encountered some difficulties managing the technology. Actions on the VR system that required a long press of the button on the controller and the lack of contrast between the screen and the pointer made clicking on items difficult. However, the participant and their caregiver did not report any falls or risks while engaging in the training program. Our preliminary results also constitute a first step in testing the combination of at-home cognitive and physical training as a possible efficient approach. The literature regarding this combination in institutional settings shows that the combined intervention produces greater improvements in cognitive function compared to either training in a single physical exercise group or a cognitive training group [10]. An animal experiment revealed that voluntary physical exercise combined with environmental enrichment offered more benefits than physical activity or enriched environments alone [25]. In humans, the combination of cognitive and physical training has been shown to improve cognitive performance in older adults, most likely due to the different additive cognitive training and physical exercise processes [26,27]. On the one hand, intensive cognitive training resulted in neural network reconfiguration and improved perceptual and executive processing [28,29]. On the other hand, physical exercise enhances neuroplasticity in specific brain areas and, as a result, improves cognitive functioning [9,30]. Therefore, cognitive and physical training may aid in improving cognitive function and combining them may result in the mutual enhancement of cognitive performance. The literature presents the outcomes of institution- and/or lab-based interventions. More evidence is needed to support the feasibility and benefits of combining cognitive and physical training administered as a semi-supervised telerehabilitation program, like the program presented in the current paper.

Additionally, the VR controller’s shape and the button’s location presented the participant with difficulties in holding it with the affected hand. In consultation with the principal investigator, the caregiver added adaptations to the right-hand controller for the VR setup to make it more functional, since the participant’s high tone created difficulty in completing the VR sessions without the caregiver’s help. One adaptation involved using an aluminum plate to hide the touchpad so that only the hand trigger on the controller’s bottom would receive input from the participant’s index finger (Figure 3). We also added adhesive Velcro to attach a brace to the controller to allow the participant to hold the controller more comfortably and functionally. The brace was used for some sessions only and then withdrawn. However, our participant still needed the caregiver to support her with clicking, highlighting the need for hand controllers adapted for people with neurologic conditions to increase the program’s feasibility. Including the family caregiver’s perspective was highly informative in this study and provided high value to our mixed-method data collection strategy, as the caregiver played a crucial role in all phases of the post-stroke rehabilitation process. Future trials should look for better options for controllers or suggest the design of neurological-condition-friendly controllers.

Motor learning has been shown to be optimized through immediate feedback and the grading of activities [31]. The functions within the VR system and tablet exercises, respectively, provided feedback on and grading of activities, but those needed improvement. The participant reported that some form of feedback when activities were completed in the VR system would have been beneficial to alert them that they successfully completed the steps required to finish the activity. Understanding that memory is a cognitive function that is targeted for rehabilitation, feedback should be graded to stop after the participant become familiar with how their movements impact the actions within the VR system to work on memory processing. Additionally, there was limited grading for fine motor exercises within the tablet sessions, which made progressing to each new session challenging, as their fine motor skills had not improved to the degree the subsequent exercises required. Similarly, tablet exercises that required the use or movement of the affected side did not account for any grading before increasing difficulty. Those aspects could be implemented in the next iteration of the prototyped technology. The results should be understood in the context of a one-case study where the intervention is presented as an adjunctive therapy. The primary focus of this paper is to report on the feasibility and safety of administering a semi-supervised telerehabilitation intervention entirely remotely. It is worth emphasizing that our participant received care from other therapists. The researchers could not gather reliable information about the type of care, frequency, and overall quality, and therefore decided to administer the proposed program as a complementary program. Now that the researchers have gathered preliminary data on the program’s feasibility, safety, and efficacy, further research should evaluate the telerehabilitation program as a standalone intervention.

The VR system presented some difficulties regarding a lack of hand function, comfortability, and technical aspects, but it was overall well received by the participant. Each piece of technology presented its own set of barriers and limitations. Our participant attended the required training session before the technology was installed in their home. Still, despite the completion of the VR training, a lack of comfort and a sense of anxiety surrounding the use of the VR system served as a barrier to using the VR system at home. A patient with a stroke is naturally determined to recover bodily functions and eager to return to pre-injury occupations. The determination of our participant and her caregiver helped them to overcome the technical difficulties encountered. For example, the couple assigned all the tasks of monitoring the laptop to the caregiver. The couple also decided to co-practice together and slowly in the beginning, especially for the task of teleportation, as it was initially very difficult for the user as they need to raise hand with the controller while holding a button to move from their current location to the next one. This movement requires a large range of motion of the shoulder, which a hemiplegic patient cannot easily achieve without compensations. Consistently in the literature, technological literacy has been discussed as a barrier to engaging in telerehabilitation amongst older adults using tablets. Vaportzis et al. (2017) [32] found a lack of knowledge and confidence and the presence of health conditions to be barriers to typical aging older adults who interact with tablets, a finding which is consistent with this study. This was not the case for our participant and the family caregiver who were younger, educated, have English as their first language, and are familiar with many technologies including cellphones and video games, which helped in the overall VR learning curve. The couple appropriately used our shared files, including a written program and a detailed explanation of the virtual environment and how to use the material. The caregiver emailed the research teams any time they had a question throughout their participation in our program. They also engaged in working with us through online shared files and evaluative online surveys without any need for detailed explanations. Our pilot study showed that technology and technological literacy can be facilitators of telerehabilitation if the patient is comfortable and eager to learn to use the technology. In our case, our virtual environments provided an immersive and goal-oriented experience that was deemed engaging and enjoyable, which increased the participant’s motivation to complete the program, in agreement with the literature [33]. However, the same program would face more barriers related to the use of technology with users who are technologically illiterate; in which case, more detailed and personalized orientation sessions would be needed before even engaging in the first virtual session.

## 5. Strengths and Limitations

There is a significant need for stroke rehabilitation services that eliminate the current barriers to accessibility faced by rural community dwellers [4,5], and which may help in the event of extreme travel and in-person meeting restrictions, such as those we lived under during the COVID-19 pandemic. This project demonstrates the feasibility and safety of integrating a VR and tablet-based TR program into the homes of stroke survivors who live rurally. While most TR research has been conducted on stroke survivors with mild physical impairments [33], the severity of one of our participant’s hemiparesis demonstrates that, with caregiver assistance, both VR and tablet TR can be used in conjunction with the usual care methods for those with moderate-to-severe physical impairments resulting from strokes. Our participant also demonstrated strong adherence to the intervention program, as was monitored by progress reports from the VR and tablet technology. Home-based interventions provide participants with the convenience of being able to initiate and complete the exercise or task-based program when it is most suitable for them. However, this unsupervised method of rehabilitation can compromise patients’ willingness to adhere to the prescribed treatment [34]. The participant was receiving weekly rehabilitation services in addition to this at-home program. This additional rehabilitation presents as a confounding parameter to any physical, cognitive, and/or emotional changes that were measured. Therefore, the outcomes observed in this study cannot be considered as the sole effects of our program but constitute the sum of the conventional supervised home training combined with our TR program. Additionally, future work should determine the outcome measures that would, realistically, be appropriate to use in a larger-scale study or implementation trials.

## 6. Future Research

Although cost-effectiveness was not evaluated in this study, it is an important aspect to consider with TR, as it has the potential to be a considerable limitation. Lloréns et al. (2015) [35] found that a VR program for balance following stroke recovery could save the participant CAD 654.72 compared to traditional in-clinic rehabilitation. Similarly, Housely et al. (2016) [36] found a cost–benefit of approximately CAD 2352.00 when using a home-based robot telerehabilitation program. However, there is a lack of additional research to support these findings. There are a number of variables that need to be considered when comparing the cost-effectiveness of TR to traditional in-clinic care. The initial start-up fees and the cost of the technology itself, the duration of time the clinician is required to spend on each session, and the distance and costs associated with travel to and from in-clinic care are all factors that need to be considered when determining cost-effectiveness [37]. Funding is also a limitation to introducing TR into homes as there is currently no private or public insurance available to cover these costs. Changes that would support TR becoming a billable means of rehabilitation need to happen on the meso- and macro-systems levels before the technology is implemented as viable rehabilitation options to increase accessibility for chronic or lifelong conditions [38]. Suggestions include advocating for the technology to be an available service in hospitals and for insurance coverage or subsidized coverage.

## 7. Conclusions

This study demonstrates that, in conjunction with traditional rehabilitation services, the use of VR and tablet-based TR is feasible in certain circumstances. The lack of comfortability with technology itself amongst the typical stroke population demonstrates the need for a more in-depth training process to ensure that individuals can manage the technology independently and safely once they are at home. Caregivers who are supporting the individuals at home should also attend and participate in the training process, as we learned that reliance on caregiver assistance with both the VR and the tablet technology was an important factor during this study. The degree of physical impairment following a stroke needs to be considered when using both the VR and tablet systems. The use of two controllers to navigate and manipulate items in the VR simulation limits the use of the technology. Although pain was noted while performing some of the tablet exercises, by trusting that our participants are experts in their own bodies and conditions, this program offered them the opportunity to skip the exercises that they felt were problematic for them.

## Figures and Tables

**Figure 1 jpm-13-01230-f001:**
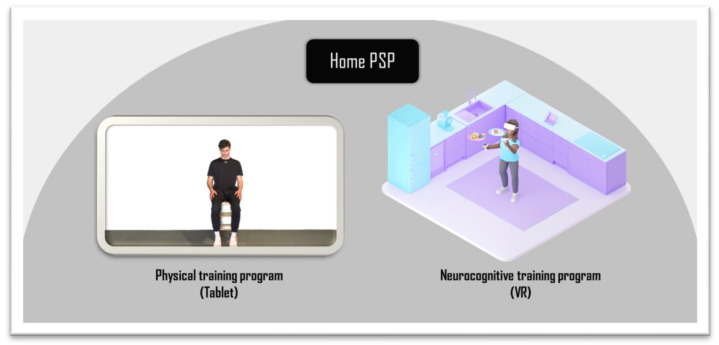
The “Active at Home Post-Stroke Program” (Home PSP).

**Figure 3 jpm-13-01230-f003:**
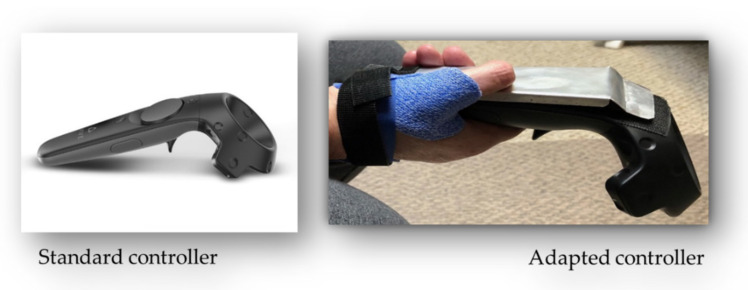
Adaptation to the HTC Vive pro controller.

## Data Availability

Data available on request due to privacy restrictions.

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
