# Peer review of "Feasibility of a Virtual-Reality-Enabled At-Home Telerehabilitation Program for Stroke Survivors: A Case Study"

_jpm, 2023, doi:10.3390/jpm13081230_

Round 1

Reviewer 1 Report

The manuscript describes an interesting telerehabilitation program using VR. New technologies and VR are indeed promising approaches for telerehabilitation in patients who live far away from therapeutic rehabilitation facilities. 

However, the manuscript is more a  description of concept in the field of VR rehabilitation than a single patient study. The study is neither a feasibility study, since the results and evaluation of N=1 is not very sounding. 

The authors should add other patients to their study.

Reviewer 2 Report

Issue is of extreme importance and the proposed program is excellent. 

However, there are some issues in the presentation that need to be adressed in my opinion.

There are two sides of the paper - the telerehabilitation idea, on which I have no major comments, and the actual rehabilitation approach, on which I will mostly comment. Approaching both of them at once does no service to either in particular. I understand this paper as a test for ONE PARTICULAR APPROACH to be used in telerehabilitation, and NOT as a telerehabilitation validation in general.  While I agree and strongly support the general setting of a home rehab system and ideea ot telerehabilitation (and HOW they do it), I am not at all convinced that the method (devices) is of any use (WHITH WHAT they do it). Of course, this is a case presentation, and the authors  clearly state that no generalization can be made, but I feel that better explanation and discussion of the rehabilitation method might add to the value of the whole.

- no explicit description of the suitable patients is given (something like "we target low impairment patients who are already able to walk and use their upper limb and do not have any orthostatic hipotension" - this should be adressed both in the methods part and in the discussions part - it also would be good to discuss how to extend the range of the method in terms of eligible patients (new exercises, new controllers, etc)

- in my opinion a biweekly training program is insufficient; as it shows later in the article, even this intensity proves hard to attain due to low motivation of the patient (having other issues to address and being tired are common excuses of patients even in the hospital setting...). In the introduction authors comment on "not enough repetitions", but go for a "mildly intensive" procedure. A comment on how much effort is needed and why they chose this intensity would be important. How to motivate the patient is also important - they do this by daily e-mails (more comments in the file)

- absolutely no information about accidents and risks is given, apart from quotings from the patient about "being tired" and comments about shoulder pain. Has the patient fallen during exercises? Is there any risk for that? Does one need supervision (not in terms of help to complete an exercise but in terms of preventing accidents)?

- device and exercises are not described as in depth as I feel necessary; accent falls on cognitive rehabilitation, while the VR physical part is quicly expedited. Choice of technical solutions (especially the use of harware controllers when systems with other types of trackers are available) are not properly discussed and explained and should be adressed (including the future research chapter..)

- the "week6...week12" approach in the text has its merrits, but in my opinion results should be systematised and shortened

- most of the "family caregiver's attitude" segment focuses on general issues (more comment on that in the file) that are only IN PART adressed by the method. I consider that most of that part should be omitted, and only the procedure-relevant areas should be maintained and properly discussed in relation to the direct advantages brought by the described approach (mostly the lack of explanation and psychological support - real issues but not relevant in my opinion unless a dedicated aspect of the method is developped)

- most of the bibliographic sources (and effort) goes towards the "telerehabilitation" issue. Since validating this article means also validating the procedure, I consider that more should be discussed about VR in terms of technical and efficacy parameters. For me the message the article sends is that of an already validated method to be used in an unvalidated setting, WHICH IS NOT THE CASE. So they should invest more to support the idea of VR (and cognitive rehabilitation) as a possible efficient approach (a plethora of contemporary attempts is available).

- a lot of testing is done during the study (great thing, but is it to be reproduced in the future studies and during the eventual clinical use after validation?). Maybe a comment on the choice of evaluation measures for a larger scale study in light of what was found here. 

        - the MMS comment in the article is funny - MMS 26 suggesting dementia after an initial MMS of 30. At least a comment on that!!! Is it related to the exercise? maybe exercising too much led to a new lacunar stroke, opening the way for a much larger debate on how much unsupervised exercise can a stroke patient do. Or maybe the "new stroke", leading to dementia, explains why the patient skipped some of the sessions. More, does a normal MMS justify the participation in a cognitive rehabilitation program (that seemingly is not needed...)?. Leaving joke aside, I do not see any reason to test cognition with MMS (it has quite a low sensibility for mild cognitive disorders) to evaluate the efficacy of a mild cognitive rehabilitation program. It can be used to adapt the set of exercises or to qualify/disqualify from the program. I do not think it worths repeating during the study. Maybe in-program scores could be used to monitor efficacy of therapy.

My comments are to be found either both in the reviewed file and in the comments section either in just one of the variants. 

I think that the base of this article is very valuable and it worths being published, but I think that it would benefit from a slight change of perspective and rewriting/adressing some issues.

Reviewer 3 Report

This article aims to explore possibilities of remote, technology-enabled home based rehabilitation program for stroke survivors.

The topic selected by authors is incredibly significant because of the of increasing number of stroke patients all over the world, their need of long-lasting rehabilitation, and on the other side raising number of obstacles for their face-to face rehabilitation program due to COVID-19 pandemic situation for example, remoteness, costs, co-morbidities, family and social issues.   

In the Introduction authors underlined the significance of timing, duration and task-oriented rehabilitation of stroke patients, and present potential advantages of virtual, tele-rehabilitation. All statements are supported by adequate literature findings.

The Aim of this study is clearly defined and presented.

Material and Methods

Intervention protocol consisting of physical training program and 5 modules neurocognitive training program is presented clearly with all details giving us complete insight in patients tasks and explaining potential benefits for patients.

Study variables are well chosen according to diagnose and status of patient.

Participant, patient, was introduced at the end of this part, my suggestion is to be introduced with patient first, then inclusion and exclusion criteria, interventional program and study variables.  

Results are presented consistently following study variables in timeline, taking into consideration point of view of patient, caregiver and objective parameters.

Discussion pointed out Strengths and limitations as well as tasks for future research.  

Conclusion give as clear opinion about place of virtual and tele-rehabilitation at this moment.

Round 2

Reviewer 1 Report

Unfortunately, I see no significant improvement of the manuscript